# Choosing where to give birth: Factors influencing migrant women's decision making in two regions of Thailand

Naomi Tschirhart[1]*, Wichuda Jiraporncharoen[2]*, Chaisiri Angkurawaranon[2], Ahmar Hashmi[2], Suphak Nosten[3], Rose McGready[4], Trygve Ottersen[1,5]

1 Department of Community Medicine and Global Health and Centre for Global Health, Oslo Group on Global Health Policy, Institute of Health and Society, Faculty of Medicine, University of Oslo, Oslo, Norway, 2 Department of Family Medicine, Chiang Mai University, Chiang Mai, Thailand, 3 Borderland Health Foundation, Thailand, 4 Centre for Tropical Medicine and Global Health, Nuffield Department of Medicine, University of Oxford, Oxford, United Kingdom, 5 Division for Health Services, Norwegian Institute of Public Health, Oslo, Norway

* wichudaj131@gmail.com (WJ); ntschirhart@gmail.com (NT)

**Data Availability Statement:** Our data is qualitative and contains personally identifiable information. Making this data publicly available would

## Abstract

### Background

Choosing where to give birth can be a matter of life and death for both mother and child. Migrants, registered or unregistered, may face different choices and challenges than non-migrants. Despite this, previous research on the factors migrant women consider when deciding where to give birth is very limited. This paper addresses this gap by examining women's decision making in a respective border and urban locale in Thailand.

### Methods

We held focus group discussions [13] with 72 non-Thai pregnant migrant women at non-government clinics in a rural border area and at two hospitals in Chiang Mai, a large city in Northern Thailand in 2018. We asked women where they will go to give birth and to explain the factors that influenced their decision.

### Results

Women identified getting the relevant documentation necessary to register their child's birth, safe birth and medical advice/quality care, as the top three factors that influenced their care seeking decision making. Language of service, free or low cost care, language of services, proximity to home, and limited alternate options for care were also identified as important considerations.

### Conclusion

Understanding factors that migrant women value when choosing where to deliver can help health care providers to create services that are responsive to migrants' preferences and encourage provision of relevant information which may influence patient decision making.

compromise participant confidentiality and privacy. Sharing excerpts of transcripts would also violate the agreement to which patients and key informants consented when they agreed to participate. For these reasons data can not be shared. Our protocol, and the associated data sharing restrictions, was reviewed and approved by Research Ethics Committee 2 at the Faculty of Medicine, Chiang Mai University (FAM-2560-05204) and was reviewed and approved through the standard internal process at the Department of Community Medicine and Global Health, University of Oslo. We have also notified our project to the Data Protection Official for Research at NSD - Norwegian Centre for Research Data (58542).

**Funding:** We acknowledge the funding which made this research possible. We received funds from the Faculty of Medicine, Department of Family Medicine, Chiang Mai University, the Life Sciences Internationalization Fund at the University of Oslo and the European Union Seventh Framework Programme (FP7-PEOPLE-2013-COFUND) under grant agreement n° 609020 - Scientia Fellows. The funders had no role in study design, data collection and analysis, decision to publish, or preparation of the manuscript.

**Competing interests:** The authors have declared that no competing interests exist.

The desire to obtain birth documentation for their child appears to be important for migrants who understand the importance of personal documentation for the lives of their children. Healthcare institutions may wish to introduce processes to facilitate obtaining documentation for pregnant migrant women and their newborns.

## Introduction

Choosing where to give birth is a multifactorial decision. Awareness of available options, individual preferences for health services, prior birth experiences, perspectives of family and friends, recommendations from health professionals and concerns about safety influence the selection of birth locations [1]. Geographic proximity also plays an important role in women's decisions on where to give birth [2]. In addition to these considerations, pregnant migrant women experience barriers to healthcare which constrain their decision making. Entitlement to publicly funded free or low cost maternal healthcare is often linked to administrative status and pregnant unregistered migrants who don't have the appropriate documentation may not be eligible for subsidized healthcare. Language barriers and associated communication difficulties pose difficulties for migrant women and there is a need to develop culturally sensitive birth support programs for these populations [3]. Studies from high income countries have found that migrant women are less likely to have sufficient prenatal care and often have higher rates of maternal mortality and morbidity than host populations [3,4]. While barriers to migrant maternal healthcare have been well documented, migrant women's decision making in relation to healthcare utilization is often overlooked in the literature. This study addresses the gap by providing the perspectives of rural and urban migrant women in the middle income country of Thailand surrounding selection of a location to give birth.

### Research setting overview

**Health coverage for migrants.** In this study we define a migrant as "an individual who has resided in a foreign country for more than 1 month or who has crossed a national border to access essential services, irrespective of the causes, voluntary or involuntary and the means, regular or irregular used to migrate" [5]. Persons who have been granted refugee status are not considered migrants. Our study is specific to international non-Thai migrants who do not have citizenship in Thailand. Migrants exist in Thailand with both regularized and irregular administrative statuses and throughout this article we use the terms documented and undocumented to reflect these respective classifications.

There are approximately 4.9 million migrants living in Thailand of which 3.1 million are from nearby Cambodia, Laos, Myanmar and Vietnam (CLMV) [6]. Thailand has made great progress toward Universal Health Coverage (UHC) and offers migrants coverage through two mechanisms: the Social Security Scheme (SSS) and the Migrant Health Insurance Scheme (MHIS). Documented migrants in the formal economy can utilize the SSS which is funded by contributions from the employee, employer and government [6]. The MHIS is designed to provide coverage for migrants who are not eligible under the SSS and is funded solely by contributions from migrants who pay 1600 baht a year (51 USD) for coverage in addition to 500 baht for an initial health examination [6,7]. Migrants must enroll initially for a two year period and thus pay 3200 baht (102 USD)[8]. MHIS is compulsory but there isn't a mechanism to ensure that all undocumented migrants participate. After 2014, migrant services were centralized and migrants were required to simultaneously register with the Ministry of Interior for a

work permit, go through national verification and get a work permit from the Ministry of Labour in order to purchase insurance coverage from the MHIS [7]. Essentially, there was a policy shift whereby undocumented migrants needed to become documented and regularized to be able to join the MHIS. Just over half (64%) of CLMV migrants who are eligible to receive coverage through these two mechanisms have enrolled and can access antenatal and birth services through the Thai healthcare system with the same coverage as Thai citizens [6]. However, it is estimated that there are an additional 800,000 undocumented migrants who are presumed to be uninsured but would be theoretically eligible for MHIS [6]. Without insurance, undocumented migrants would receive birth care if they present at a Thai hospital while in labour but would be billed for the services.

To address the persistent need for healthcare among migrant communities along the Thailand-Myanmar border, the Migrant Fund (M-FUND), a non-profit insurance scheme, was developed to provide health coverage for undocumented migrants. Initially launched with monthly premiums of 60–150 baht (2–5 USD) for M-FUND version 1, at the time that we wrote this manuscript the M-FUND was in version 3[8]. We conducted this research in 2018 just prior to the transition to M-FUND version 2 and at that time research pregnant women were predominantly enrolling in the 100 baht plan which covered hospital admissions, and outpatient consultation to a maximum coverage of 60,000 THB/ year (1964 USD). To date over nine thousand migrants are registered in the M-FUND and are able to receive healthcare for a broad range of conditions including pregnancy at five collaborating hospitals and clinics [9].

**Border area and Chiang Mai city.** This research took place in two research locales in Thailand, one along the border with Myanmar and a second in a peri-urban area of a large city.

Tak province, Thailand shares a long and porous border with Myanmar and has a long history of migrant and refugee healthcare provision by non-profit organizations. As Myanmar has opened up, donor funding has moved into the country and further investments have been made into maternal health provision at government hospitals. In addition, refugee camps on the Thai side are in the process of closing. These changes along the border have led to a reduction in funds for non-profit health organizations. However, keeping with a thirty year tradition of low-cost healthcare provision, these organizations continue to provide free or low cost antenatal and delivery services to undocumented migrants who are not covered by the Thai government's SSS or MHIS schemes. Migrants with insurance coverage can of course also access the Thai public healthcare system in Tak province.

Chiang Mai is the second largest city in Thailand and is situated in the country's northwestern region. Public healthcare in and around Chiang Mai is provided by the Thai public healthcare system and this region does not have a history of non-profit migrant specific health clinics as found in Tak province. In both the border region and Chiang Mai, uninsured persons can utilize Thai government hospital care but will be billed for the services.

## Methods

In designing the study we opted to utilize focus groups as this method helps to facilitate conversations between participants, is culturally appropriate and has been successfully utilized by other studies in this setting [10–12]. We held focus group discussions [13] with 72 non-Thai pregnant migrant women at three non-government clinics in a rural border area and at two hospitals in Chiang Mai, the largest city in Northern Thailand in May and June of 2018. We recruited pregnant women who self-identified as non-Thai migrants and did not have an acute medical or obstetrical condition that would be disrupted or delayed by involvement in focus

group discussion. Along the Thailand-Myanmar border we sought individuals who were fluent in spoken Burmese or Karen and in Chiang Mai individuals who spoke sufficient Burmese, Karen or Thai language to be able to participate in the research. Where possible, we held separate group discussions for first-time and experienced pregnant women. All focus groups were held at health facilities in a quiet location, separate from where care was being provided. At the beginning of each discussion, the facilitator and interpreter read the consent form out loud. Participants provided verbal consent and the facilitator signed the consent form to document that consent had been obtained. In the border area NT, a researcher with training in qualitative methods, lead the discussions and SN, a skilled community based researcher, provided interpretation to help facilitate the dialogue. NT speaks English, French and Thai and previously conducted her doctoral fieldwork in Thailand. SN is fluent in English, French, Thai, Karen and Burmese and has provided interpretation and facilitation support for multiple community based studies in the research setting. In Chiang Mai, Kulyapa Yoonut lead the focus group discussions in Thai and SN provided primary interpretation with support for Shan language from an undergraduate university student. Ms. Yoonut is a Thai study nurse with focus group expertise. Members of the research team including WJ, CA, AH and a study assistant were present for some of the group discussions.

Prior to the fieldwork, we developed a series of images with examples of possible reasons why women would seek to deliver birth at a specific location. We decided to use photos and drawings produced by the research team to engage participants, many of whom had low literacy levels, as these methods have been used to engage participants in prior studies [10,11]. These images were based on themes identified in the literature and previous informal discussions with migrant health care providers. We piloted the cards and images with members of the research team and colleagues who are from Myanmar to ensure that the meanings of the images and terms were appropriately translated and culturally appropriate. The themes in our photo cards included: safe birth, birth certificate, close place, medical advice, friend or family advice, only option, free, fear, previous problem and language.

We utilized convenience sampling and where possible we attempted to create focus groups that included women similar in gravidity (primi and multi), language and ethnicity. Healthcare workers informed potential participants about the study and invited interested persons to meet at a specific time and location to join the discussion. In each focus group, we utilized a semi-structured interview guide and asked women where they will go to give birth and to explain the factors that influenced their decision. We introduced the photo cards one- by-one, explaining the theme and asked participants to select the reasons that were important to them and to place them in a pile. We then asked them to choose their top three reasons and discuss. We also asked women to identify any additional considerations which influenced their choice of birth locale. In addition to audio recording the sessions, two members of the research team also took field notes.

Following the focus group discussions, all of the audio files were subsequently transcribed and translated into English by an independent translator. In the case where we needed to clarify meaning or linguistic nuances members of the research team fluent in languages used by participants reviewed segments of the audio file. We uploaded the files into NVivo (version 11) qualitative software and conducted thematic analysis to organize the data around pre-identified and emergent themes related to considerations when choosing a place to give birth. Using a coding frame initially comprised of the pre-identified themes NT coded the data while continuously updating the frame to include emergent themes. Team members subsequently met to discuss the analysis. For the photocard activity, we documented the factors women considered important when selecting care by giving each factor a point from every focus group where it was prioritized. We identified the priorities of the women within each focus group,

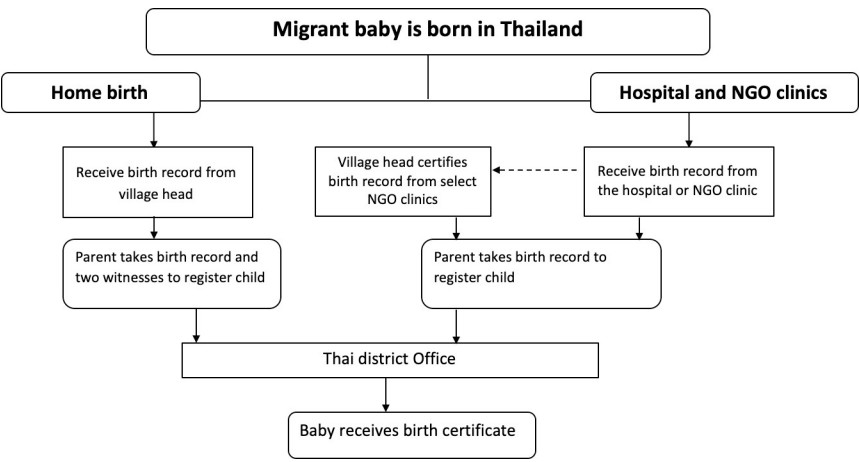

**Fig 1. Pathways to receive a birth certificate for migrant children born in Thailand.**

and then enumerated the frequency with which women selected these important factors across all groups. Similarly, we report the top priorities determined by each respective focus group.

In addition to the focus groups, we also conducted key informant interviews [18] with maternal health providers who worked with migrant populations in both the border and city region. Based on the data obtained from both focus group participants and key informants we developed a diagram of birth registration pathways (Fig 1). To further triangulate the data we verified our preliminary results with several key informants that we had interviewed as well as additional stakeholders knowledgeable about birth registration.

The research project was reviewed and approved by Research Ethics Committee 2 at the Faculty of Medicine, Chiang Mai University (FAM-2560-05204). The project was also reviewed and approved through the standard internal process at the Department of Community Medicine and Global Health at the University of Oslo and notified to the Data Protection Official for Research at NSD—Norwegian Centre for Research Data (58542). In addition, we sought input from the Tak Province Community Advisory Board.

## Results

### Description of the study sample

We held a total of thirteen focus group discussions (72 women) including nine at border clinics on the Thai side of the border in Tak province and four at hospitals in peri-urban Chiang Mai. Participants along the border spoke Burmese, Karen (Sgaw and Pwo dialects), while in Chiang Mai their first language was Shan and Chin. We present participant characteristics in Table 1. All were migrants from Myanmar and most were experienced mothers. Integration

**Table 1. Characteristics of focus group participants (n = 72).**

| Characteristics | N |
|---|---|
| Location of Antenatal care | |
| Border sites | 54 |
| Chiang Mai hospitals | 18 |
| Number of pregnancies | |
| 1 | 27 |
| >1 | 45 |

into the formal Thai economy and associated health insurance coverage varied between the study sites with participants from Chiang Mai more likely to have Thai government SSS or MHIS coverage. Many of the women in the border clinics were participating in the M-FUND, a non-profit health insurance program specifically for undocumented migrants.

## Planned birth location

Most women expressed that they planned to give birth at the clinic or hospital where they were currently receiving antenatal care. Participants could access care at these locations for free or 30 baht (1USD). For some, especially in the border area, the clinic where they were seeking antenatal care was the only one they knew about where they could get low cost services. Their other low cost option would be to give birth in the village with the assistance of a traditional birth attendant. Hospital births in Myawaddy (Myanmar) and Mae Sot Hospital (Thailand) were regarded as expensive by undocumented women.

Women in peri-urban Chiang Mai were much more knowledgeable about the various hospitals where they could give birth, but preference for birth location was somewhat pre-determined, as women who signed up for the MHIS program received a designated hospital where they could receive low cost care. For care to be covered at another hospital they needed to receive a referral.

## Important considerations when deciding where to give birth

We did not find a notable difference in the top three considerations between focus groups irrespective of differences in ethnicity, language, gravidity or rural versus urban setting.

Women indicated that the top three most important considerations when selecting a birth location were: receiving the necessary paperwork to secure a birth certificate, having a safe birth with good health outcomes, and receiving medical advice and quality care.

**Birth certificate.** When giving birth in a clinic or hospital in Thailand, migrants receive paperwork that they can then use to register the birth with local Thai authorities and to ultimately obtain a birth certificate for their child. During the focus group discussions in both study locations, women highly valued getting a birth certificate. Having a Thai birth certificate comes along with certain entitlements from the Thai government including access to UHC healthcare for the child and ability to enroll in government schools. A birth certificate, as a form of documentation, would also permit greater mobility and freedom of movement in Thailand and between Thailand and Myanmar. Women explained that the certificate could be used for the child's whole life and would help them "not to suffer like their parents".

Having a birth certificate was equally valued by women who planned to raise their children in Myanmar as they believed it was necessary to register children for school. Women described it as more robust than the census registration where names are added to the family house registry. One woman explained that she had successfully registered her child in a Myanmar school without the birth certificate but she could not transfer them to another school and another indicated that without these documents they could not move their children across the border to Thailand.

Many of the participants had experiences living in Thailand without the proper documentation, and this also impeded their ability to register their child's birth and to obtain their birth certificate. One woman explained that her daughter was born in Thailand but she was not able to obtain a birth certificate.

*I gave birth here but I didn't get a birth certificate. When I came to give birth here, I had my passport. But later, I lost my passport. So I didn't have a passport after I delivered. The health*

*worker asked me to bring my passport. Come back and get it on the 21st. I couldn't get my daughter's birth certificate because I lost my passport.*

Fig 1 shows the pathways for parents to obtain a birth certificate for their child in Thailand. Women expressed that it was easier to obtain the necessary paperwork to register their child's birth if they gave birth in a clinic or hospital as compared to the village. Some indicated that registration in the village would be costly and time consuming. A participant explained,

*For my first child, it took almost two years to get a birth certificate. We have to get it after we deliver. When you live in the village you have to go through different steps. But now they ask us to go and deliver at the hospital so that they can provide us with the birth certificate right away.*

Stakeholders indicated that parents whose child was born at home, have to take the additional step of bringing two witnesses to the district office to register the child which would add time and complexity to the process.

Several maternal care providers in the border area, have teams which provide information on birth registration to pregnant women including a list of the required documents and then assists with transportation after delivery to the district office for birth registration. Stakeholders explained that birth certification in Thailand is available regardless of whether a migrant has documents or not and thus irregular or undocumented migrants are also eligible to receive a birth certificate for their child.

For women who give birth in Thailand and wish to receive a birth certificate from Myanmar, Myanmar policy does not automatically recognize Thai birth certificates. Instead the parents must present to their local village in Myanmar and begin the process or register their child with the Myanmar embassy in the country the child was born within one month of birth. Children whose birth is registered in Myanmar at the village level and who subsequently receive a Myanmar birth certificate will be officially "born" in Myanmar despite having been actually birthed in Thailand.

In Thailand, parents need to present to get a birth certificate within 15 days of birth. According to participants, going back and getting a child's birth certificate years after their birth was a difficult and costly endeavor. One woman noticed two children at the factory where she works who don't have birth certificates. She perceived that they would have to pay a large sum of money to get their documentation processed,

*The children are getting bigger at the age to attend school. If they do the birth certificate, it will cost 30,000 (baht) (990 USD) for each. If they have the birth certificate, their names will be added to the household register. After that, they will be accepted at the school. They were born on this side so they don't have a birth certificate from the Myanmar side.*

Obtaining a replacement birth certificate was also costly and one woman had to pay 50,000 kyat (33 USD) to have it reissued. She explained, "For the very first time after our child was born, we didn't have to pay. But if we lost it afterwards, we'll need to pay. When we lost it, it seems like we don't respect them".

**Safe birth.** Having a safe birth with positive health outcomes for their child helped motivate women to give birth in the clinic or hospital. One woman described her reasoning, "This is both us and our child. We are a lot safer from the danger of delivering when we come to deliver at the clinic". Participants expressed that as parents they would be happy if their children are healthy.

**Medical advice and quality care.**    Participants expanded the concept of medical advice, from advice received during care, to include quality care. Receiving quality healthcare from health professionals with medical expertise in handling birth complications were identified by participants as important considerations when deciding where to give birth. A woman explained, "When we're at the hospital (clinic) whether or not we have a cesarean, we have quality care from health workers here at the clinic. So that's better than to deliver at home". Participants were confident that the health workers at the clinic or hospital would be able to take care of their child whether it was healthy or not.

In addition to the top factors women identified, receiving low or no cost care, language, proximity, and the clinic as their only option were other important considerations when deciding where to give birth. Very few groups indicated that past problems, advice from friends or family and fear were important in selecting a place to give birth. There were some subtle differences in women's considerations from the rural border and urban settings.

**Affordable care.**    In the border region, women who attended antenatal care at the non-profit clinics were able to access services for uncomplicated births free of charge. The M-FUND, a non-profit health insurance fund, was also put into place where women could make monthly contributions and have access to services at a larger hospital if necessary. Participants expressed that care affordability was an important factor when choosing where to give birth. In one woman's words, "It will be difficult for us if we don't have enough money when we get labor pain". Fees for birth services at the nearby Thai and Myanmar government hospital were considered high for undocumented participants. However, some women felt that free or low cost services were not as important as the others as "It's something we need to pay for if we really need it".

In Chiang Mai, most of the participants were enrolled in the MHIS program and were planning to give birth at a designated hospital where they would only have to pay 30 baht (1USD) to give birth. While knowledgeable about other options, they chose to birth at their designated hospital instead of paying more to go to another hospital.

**Language.**    The importance of language when deciding where to give birth varied among focus groups. Women in the Chiang Mai urban setting could often speak and understand Thai and thus did not see language as an important consideration. Additionally, many of them spoke Shan and to our knowledge birth services in Shan language are not offered in Thailand. They also valued doctors and nurses who were easy to communicate with and spoke to them using simplified Thai language.

For women on the border area, being able to receive care in Karen or Burmese was helpful. However, some indicated that they would bring someone to interpret if they went to a Thai hospital.

**Close by.**    Proximity to healthcare services was identified by some participants as a consideration when choosing where to seek care. Getting care close by was valued by women and they used travel time to approximate distance from their home to the healthcare facility. Participants had different definitions of closeness. For example, one woman who was living in a village in Myanmar spent two hours walking to reach the clinic in Thailand and perceived the clinic as close to her home. She explained, "We need to consider it as close for us". In the Chiang Mai urban setting, many of the women lived near the hospital where they were registered.

**Only option.**    Participants in the clinics along the Thailand-Myanmar border indicated that the non-profit clinic where they planned to give birth was the only healthcare option that they knew about and which they saw as accessible to them. Contrastingly, none of the urban participants perceived their designate Thai hospital as the only choice.

## Discussion

This is the first study to assess factors in choosing a place to deliver among migrants as far as we are aware. The three most important considerations for migrant women when considering where to give birth were securing paperwork necessary to obtain a birth certificate, experiencing a safe birth and receiving medical advice and quality care. These three factors were consistent among both border and urban migrant participants.

### Choice

For migrant women in this study, birth places where women perceived they could access care were limited by knowledge of available clinics and hospitals and entitlement to free or low cost birth services. Some of the participants along the Thailand-Myanmar border only knew of the specific clinic where they were receiving care and viewed giving birth at home in the village with the assistance of a traditional birth attendant as their other option. Migrants in the urban Chiang Mai group were much more knowledgeable about other hospitals where they could give birth and didn't mention the possibility of home birth. The urban and border groups were very different, the former being documented, able to comprehend Thai language and registered in the Thai government's MHIS scheme which gave them access to care at a designated hospital. The rural border group had participants with a greater variety of documentation statuses including a large number who would not be eligible for low cost care at a Thai hospital.

At the time of the research, all of the women were receiving free or low-cost birth services and most planned to give birth at the same clinic or hospital where they were receiving care. Both groups of migrants were confident in the quality of care they would receive from the healthcare providers and believed that they would be able to get referred onwards for more complicated care if necessary. Most participants had insurance coverage from the government or non-profit sector which would help to prevent catastrophic costs associated with a Caesarean section at a Thai hospital for an uninsured migrant.

### Top factors

While participants' choice of place of birth must be considered in light of participants' narrow range of options, when we asked them to identify the top three considerations they came to consensus around safe birth, medical advice or quality care and documentation necessary to obtain a birth certificate. Safety and quality care have been documented internationally as key factors when seeking maternal healthcare [13]. A desire to have a safe birth facilitated through quality healthcare, including access to necessary emergency interventions and positive associated health outcomes for mother and child has universal appeal and echoes global health policy objectives [14].

However, striving to get documentation necessary to register their child's birth, is not directly linked to immediate health outcomes for woman or child, and we suspect is specific to participants' experiences as migrants. Women described their lived experience being a migrant in Thailand without the proper documentation and wanted greater stability for their children. Another study with Shan migrants in Thailand reported safe birth and acquiring identification documents as motivations for attending antenatal care [15]. Thailand does not provide citizenship to children born in Thailand of migrant parents, but the country's policies to provide elementary education and low-cost healthcare coverage to migrant children may also incentivize birth registration. According to a 2005 cabinet resolution, all "non-Thai" children in Thailand can receive education in grades 1–9 at Thai public schools [16]. While according to policy, migrant children should be able to receive free education at Thai public schools without providing proof of identity, a study from another Thai province reports teachers requesting

identification documents [16]. This requires some discussion, as women's perception related to access to services such as education and health services, may not only reflect Thai or Myanmar policy, but also local practices that can constrain access. Our participants believed that a birth certificate was necessary to register a child in school and thus it is possible that administrative barriers on the ground and the associated request for documentation create real difficulties for migrants. Low cost basic healthcare coverage can be purchased for migrant children who are dependents of migrant workers through the MHIS with yearly cost of 11 USD for children under 7 years old [16]. Migrant infants who have obtained a birth certificate can also be registered for MHIS coverage.

Birth certificates are also necessary to obtain travel documents and thus also help children to move around freely which is another incentive. We suspect that there has been a policy shift towards improved personal documentation in both Thailand and Myanmar in the last decade, as indicated by the national verification process, which has fueled the need for birth certificates on both sides of the border [17]. Revisions to the Civil Registration Act in 2008 allowing birth registration for migrant children in Thailand and a 2015 policy permitting children born before 2008 to retroactively apply for birth registration are also evidence of the movement towards improved documentation [18].

Our participants reported the importance of getting a birth certificate for their child soon after the birth and Thailand has a policy of providing free birth registration in the first fifteen days after the infant is born [19]. This short time window presents an additional difficulty for mothers who are themselves undocumented. A few women reported previously giving birth in Thailand but were not able to receive a birth certificate for their child as they themselves did not have their own personal identification documents at that time. From discussions with stakeholders working on the ground, we understand that migrant women do not need to have personal documentation to register their child's birth certificate. However, we perceive that undocumented women may feel uncomfortable presenting at the district office to request a birth certificate due to their own precarious status and an associated fear of authorities. Inability to speak Thai (or read in Karen or Burmese) may be another barrier for parents who do not have support from someone who can interpret during the registration process [10]. Women, especially first-time mothers, may also not be knowledgeable of pathways to birth registration. Being unaware of the need for birth registration or the process has been reported as barriers to registration in the Thai context [20]. Migrant children who are born in Thailand but do not receive a birth certificate are at risk of becoming stateless, meaning that they are not recognized as a citizen of any country, if their parents do not register their birth in Myanmar. Lack of birth registration contributes to a cycle of statelessness and a loss of future entitlements for migrant children.

In the literature, outside of Thailand little has been reported on birth certificates as consideration for migrants selecting a location to give birth outside of the remit of birth tourism in countries that provide birthright citizenship where infants born in the territory are immediately entitled to citizenship [21]. Given the value of registration in securing social, economic and political rights of future generations of migrants, we anticipate that birth registration is an important consideration for migrants choosing a birthplace worldwide. We urge scholars to consider the importance of registration when designing for migrant maternal healthcare research.

## Other factors

Internationally care affordability has been identified as a challenge for undocumented pregnant mothers in countries that do not provide free care to this group [22]. Women in most of

our focus groups indicated that low cost or free services were important when deciding where to give birth and at the time of the research all were receiving affordable care. They valued the provision of services in a language migrants understand and were able to comprehend the care they received in Karen, Burmese and Thai. It is possible that participants in our research did not consider affordability and language among the most pressing concerns when seeking care, as they are already able to use free or low cost services in languages they understand and thus these considerations while important were not highest on their list. Thai hospitals in the border region frequently have Karen and Burmese interpreters, although they may have large caseloads. Alternatively, many in the border region likely knew that the Non Governmental Organization (NGO) facility would provide a bridging person (cost-free) who speaks Karen/Burmese and Thai if referral to a different language facility was needed. Thailand is unique as both groups of migrant participants were able to get some insurance coverage from the government and non-profit sector respectively, which influenced their entitlements. In contrast to other settings internationally communication challenges and gaps in healthcare coverage have been documented as difficulties for pregnant migrant women seeking healthcare [3].

## Policy and practice recommendations

We anticipate that undocumented mothers may face additional challenges when attempting to register their child's birth and recognize that making it easier for pregnant women to obtain documentation prior to giving birth, may help to ease registration. NGOs and government healthcare facilities may consider adopting institutional policies that help women ensure they have the necessary documentation prior to their delivery. Registration support is resource intensive as transportation and translators are required and consideration should be given to how this can be funded. Efforts from NGOs to help interpret registration laws and assist migrant parents in navigating registration have reported successful outcomes and expanded partnerships between district offices and clinic/hospitals where women give birth could help to smooth the process of receiving birth certificates for migrant children [23].

## Strengths and limitations

Participants participated in the analysis by ranking the factors during the focus group which added to the robustness of the study. By including migrants from two different regions, with the inclusion of non-profit clinics and government hospitals, we are able to strengthen our study. In interpreting our research, a limitation is that we conducted fieldwork in 2018 and policies and implementation frequently change. With this in mind, this study should be considered within the specific time period. One limitation of our study is that we didn't speak with women who elect forgo antenatal care and those only arriving for emergency obstetric care. Further discussions with these women would present a more complete picture as they are likely to represent the more marginalized.

We anticipate that the ranking of which considerations are most important will depend on current care. As all women who participated in this research were receiving free or low-cost maternal care, affordability may not be reported as a top consideration in their decision making in this specific context, even if it would be a top consideration when asked more generally to migrants in other geographical regions.

## Conclusion

In efforts to create services that are responsive to migrants' needs, health care providers can benefit from understanding the factors which influence migrant women's decision making. This knowledge can be used to shape services and to provide patients with information they

can use when deciding where to give birth. Obtaining birth documentation for their child appears to be a factor which is specific to migrants and healthcare institutions may wish to develop processes to help obtain documentation for pregnant migrant women and migrant newborns.

## Acknowledgments

Thank you to Mae Tao Clinic, the Borderland Health Foundation, Shoklo Malaria Research Unit, Saraphi Hospital and San Sai Hospital for contributing to this project. Thank you to Kulyapa Yoonut, Patcharin Puttanusegsan and Napat Khirikoekkong for assisting with data collection. We gratefully acknowledge the contributions of our participants who shared their experiences with us and in Tak Province the border hospitals, Mae Ra Mat and Po Prah District Hospitals and Mae Sot General Hospital, for ongoing support.

## Author Contributions

**Conceptualization:** Naomi Tschirhart, Chaisiri Angkurawaranon, Rose McGready, Trygve Ottersen.

**Formal analysis:** Naomi Tschirhart.

**Investigation:** Naomi Tschirhart, Wichuda Jiraporncharoen, Chaisiri Angkurawaranon, Ahmar Hashmi, Suphak Nosten.

**Project administration:** Naomi Tschirhart, Wichuda Jiraporncharoen, Chaisiri Angkurawaranon, Trygve Ottersen.

**Validation:** Naomi Tschirhart, Wichuda Jiraporncharoen, Chaisiri Angkurawaranon, Ahmar Hashmi, Suphak Nosten.

**Writing – original draft:** Naomi Tschirhart.

**Writing – review & editing:** Naomi Tschirhart, Wichuda Jiraporncharoen, Chaisiri Angkurawaranon, Ahmar Hashmi, Suphak Nosten, Rose McGready, Trygve Ottersen.

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
