## [Decision Letter · Decision Letter 0]

9 Dec 2019

PONE-D-19-29338

Choosing where to give birth: Factors influencing migrant women’s decision making in two regions of Thailand

PLOS ONE

Dear Dr Jiraporncharoen,

Thank you for submitting your manuscript to PLOS ONE. After careful consideration, we feel that it has merit but does not fully meet PLOS ONE’s publication criteria as it currently stands. Therefore, we invite you to submit a revised version of the manuscript that addresses the points raised during the review process.

We would appreciate receiving your revised manuscript by Jan 23 2020 11:59PM. To enhance the reproducibility of your results, we recommend that if applicable you deposit your laboratory protocols in protocols.io, where a protocol can be assigned its own identifier (DOI) such that it can be cited independently in the future. For instructions see: http://journals.plos.org/plosone/s/submission-guidelines#loc-laboratory-protocols

We look forward to receiving your revised manuscript.

Kind regards,

Calistus Wilunda, DrPH

Academic Editor

PLOS ONE

Journal Requirements:

**When submitting your revision, we need you to address these additional requirements:**

**Please ensure that your manuscript meets PLOS ONE's style requirements, including those for file naming. The PLOS ONE style templates can be found at http://www.plosone.org/attachments/PLOSOne_formatting_sample_main_body.pdf and http://www.plosone.org/attachments/PLOSOne_formatting_sample_title_authors_affiliations.pdf**In your Data Availability statement, you have not specified where the minimal data set underlying the results described in your manuscript can be found. PLOS defines a study's minimal data set as the underlying data used to reach the conclusions drawn in the manuscript and any additional data required to replicate the reported study findings in their entirety. All PLOS journals require that the minimal data set be made fully available. For more information about our data policy, please see http://journals.plos.org/plosone/s/data-availability.Upon re-submitting your revised manuscript, please upload your study’s minimal underlying data set as either Supporting Information files or to a stable, public repository and include the relevant URLs, DOIs, or accession numbers within your revised cover letter. For a list of acceptable repositories, please see http://journals.plos.org/plosone/s/data-availability#loc-recommended-repositories. Any potentially identifying patient information must be fully anonymized.Important: If there are ethical or legal restrictions to sharing your data publicly, please explain these restrictions in detail. Please see our guidelines for more information on what we consider unacceptable restrictions to publicly sharing data: http://journals.plos.org/plosone/s/data-availability#loc-unacceptable-data-access-restrictions. Note that it is not acceptable for the authors to be the sole named individuals responsible for ensuring data access.We will update your Data Availability statement to reflect the information you provide in your cover letter.Please provide additional details regarding participant consent. In the ethics statement in the Methods and online submission information, please ensure that you have specified how verbal consent was documented and witnessed.Please consider discussing some of the potential bias introduced in your qualitative study.Your ethics statement must appear in the Methods section of your manuscript. If your ethics statement is written in any section besides the Methods, please move it to the Methods section and delete it from any other section. Please also ensure that your ethics statement is included in your manuscript, as the ethics section of your online submission will not be published alongside your manuscript.

Thank you for your attention to our queries.

Additional Editor Comments (if provided):

Please submit your revised manuscript alongside a COREQ checklist, or other relevant checklists listed by the Equator Network, such as the SRQR, to ensure complete reporting.

Reviewers' comments:

Reviewer's Responses to Questions

**Comments to the Author**

1. Is the manuscript technically sound, and do the data support the conclusions?

Reviewer #1: Yes

Reviewer #2: Yes

2. Has the statistical analysis been performed appropriately and rigorously? 

Reviewer #1: N/A

Reviewer #2: N/A

3. Have the authors made all data underlying the findings in their manuscript fully available?

Reviewer #1: Yes

Reviewer #2: No

4. Is the manuscript presented in an intelligible fashion and written in standard English?

Reviewer #1: Yes

Reviewer #2: Yes

5. Review Comments to the Author

Reviewer #1: This study is provided data from the interviewing subjects. The author has a good system, well defined for the methodology and provide the data referred to the situation of health care and policy of the country.

Reviewer #2: Thank you for the opportunity to review this interesting paper. There is a substantial amount of information missing from the Methods section that must be addressed, and methodological choices need to be more explicitly justified. The use of a tool such as the COREQ checklist (https://www.equator-network.org/reporting-guidelines/coreq/) might be helpful to the authors when they are writing up their Methods for qualitative papers in the future. Other than that, I have mostly minor revisions

1. Abstract: please include number of participants in Abstract.

2. Line 59: ‘local’ should be ‘locale’ or ‘location’

3. Line 94: Are you able to provide an approximate / average cost for this service?

4. Methods: Why did you choose focus group discussions to investigate this issue rather than interviews with women? What made FGDs the better choice?

5. Methods: Missing information about focus group facilitator: Who facilitated the focus group discussions? One of the authors? An independent facilitator? What was their experience / training / qualifications relevant to conducting FGDs? What were their characteristics? E.g. nationality / gender / languages spoken – could these have impacted how they interacted with participants / how participants interacted with them?

6. Methods: Missing info about interpreter: Who was the interpreter? Were they independent from the research team or part of research team? Were they known to the participants? Did they have experience / training in facilitating focus groups? What were their relevant characteristics that may have affected dynamic of focus group discussion?

7. How were participants recruited? Approached face to face? Flyers? Told about the study by health workers? Did you record if/how many refused to participate and why?

8. Did you sample purposively or use a convenience sample?

9. What’s the rationale behind your sample size?

10. Where were focus groups held? (i.e. physical location – heath facilities, community spaces, someone’s home, NGO offices?)

11. Was anyone else present during focus group discussions besides the participants and facilitators?

12. Please provide more information about your analysis process, specifically the coding. How many data coders were they? Did they develop a coding tree based on inductive analysis (in order to develop the “emergent themes” you mention in addition to your deductive themes).

13. Did participants or stakeholder provide feedback on your findings?

14. Who translated the audio files? Was this a member of the research team or an independent translator? Were they involved in the analysis? Was there any process of back translation or process to clarify meaning / identify linguistic nuances / identify cultural assumptions that may occur in the translation process?

15. Results: Much of the information in lines 180-186 would be better presented in a Table, rather than as descriptive text.

16. Line 209: The use of the words ‘significant difference’ here implies a statistical comparison has been performed. Use a synonym for significant, such as notable, meaningful, important.

17. Line 282: Please provide USD value of 30,000 baht

18. Discussion: Women in your study did not consider shared language to be a pressing concern, perhaps because they were mostly assured of being provided services in a language that they spoke, or being provided with a ‘bridging person.’ But is this in contrast or a similarity to other populations internationally? (e.g other migrant populations, indigenous and ethnic minorities) What is in the literature about choice of birth location and shared language /related concepts such as cultural safety?

6. PLOS authors have the option to publish the peer review history of their article (what does this mean?). If published, this will include your full peer review and any attached files.

Reviewer #1: No

Reviewer #2: No

---

## [Author Response · Author response to Decision Letter 0]

17 Feb 2020

January 20, 2020

Re: PONE-D-19-29338

Choosing where to give birth: Factors influencing migrant women’s decision making in two regions of Thailand

Dear Calistus Wilunda, PLOS ONE academic editor, 

Thank you for considering our manuscript. In this letter we provide a point by point response to your comments. We have also uploaded a manuscript with changes highlighted ‘Revised Manuscript with Track Changes’ and an unmarked version of the manuscript labeled ‘Manuscript’. 

-----------

Journal Requirements:

Response: We have reviewed the style requirements.

Response: There are legal restrictions, specifically participant confidentiality, which restricts us from sharing the data. 

Please update our data availability with the following statement “Our data is qualitative and contains identifying information. Data cannot be made publicly available for ethical reasons, as public availability would compromise patient confidentiality and participant privacy. Sharing excerpts of transcripts from this research would violate the agreement to which patients and key informants consented when they agreed to participate”.

We used this statement in a previous PLOS ONE publication. 

3. Please provide additional details regarding participant consent. In the ethics statement in the Methods and online submission information, please ensure that you have specified how verbal consent was documented and witnessed.

Response: We have updated the section to provide more details. See (Page 7, line 147-149), “At the beginning of each discussion, the facilitator and interpreter read the consent form out loud. Participants provided verbal consent and the facilitator signed the consent form to document that consent had been obtained”.

4. Please consider discussing some of the potential bias introduced in your qualitative study.

Response: Thanks for this query. Qualitative research often examines the nuances of rigour and trustworthiness of the research project and results in response to questions related to bias. To enhance our rigour for the study in the methods we utilized a semi-structured interview guide and more importantly asked participants to contribute to the analysis by ranking factors of importance themselves. We also sought to include migrants from different regions to enhance the robustness of our study. See page 22 lines 547-550, “Participants participated in the analysis by ranking the factors during the focus group which added to the robustness of the study. By including migrants from two different regions, with the inclusion of non-profit clinics and government hospitals, we are able to strengthen our study”. In the limitation section we also explain that our study did not include women who forwent antenatal care and only came to the hospital in an emergency, thereby contextualizing who was included and not included in the project.

We have updated the text to indicate the use of a semi-structured interview guide, (page 8, lines 176-178) “In each focus group, we utilized a semi-structured interview guide and asked women where they will go to give birth and to explain the factors that influenced their decision.”

5. Your ethics statement must appear in the Methods section of your manuscript. If your ethics statement is written in any section besides the Methods, please move it to the Methods section and delete it from any other section. Please also ensure that your ethics statement is included in your manuscript, as the ethics section of your online submission will not be published alongside your manuscript.

Response: We have moved the ethics statement to the methods section of the manuscript and have deleted it from the end of the article. See lines 210-215 on page 9.

Additional Editor Comments (if provided):

Please submit your revised manuscript alongside a COREQ checklist, or other relevant checklists listed by the Equator Network, such as the SRQR, to ensure complete reporting.

Response: We have included the elements of the COREQ checklist in our revised methods section (responding to the feedback from reviewer 2).

Comments to the Author

1. Is the manuscript technically sound, and do the data support the conclusions?

Reviewer #1: Yes

Reviewer #2: Yes

2. Has the statistical analysis been performed appropriately and rigorously? 

Reviewer #1: N/A

Reviewer #2: N/A

3. Have the authors made all data underlying the findings in their manuscript fully available?

Reviewer #1: Yes

Reviewer #2: No

Response: Due to ethical considerations, we cannot make our data publicly available. We have requested inclusion of the following statement, “Our data is qualitative and contains identifying information. Data cannot be made publicly available for ethical reasons, as public availability would compromise patient confidentiality and participant privacy. Sharing excerpts of transcripts from this research would violate the agreement to which patients and key informants consented when they agreed to participate”.

4. Is the manuscript presented in an intelligible fashion and written in standard English?

Reviewer #1: Yes

Reviewer #2: Yes

5. Review Comments to the Author

Reviewer #1: This study is provided data from the interviewing subjects. The author has a good system, well defined for the methodology and provide the data referred to the situation of health care and policy of the country.

Response: Thank you for this feedback and for taking time to review our paper.

Reviewer #2: Thank you for the opportunity to review this interesting paper. There is a substantial amount of information missing from the Methods section that must be addressed, and methodological choices need to be more explicitly justified. The use of a tool such as the COREQ checklist (https://www.equator-network.org/reporting-guidelines/coreq/) might be helpful to the authors when they are writing up their Methods for qualitative papers in the future. 

Response: We appreciate your suggestion that we consider the COREQ checklist for future papers. We have provided more information in the Methods section in response to your comments, see responses 4-14 below for more information. 

Other than that, I have mostly minor revisions

1. Abstract: please include number of participants in Abstract.

Response: Thanks for this suggestion. We have added the number on line 23 page 2.

2. Line 59: ‘local’ should be ‘locale’ or ‘location’

Response: Well received. We have updated this to “location” on page 3, line 59.

3. Line 94: Are you able to provide an approximate / average cost for this service?

Response: The cost for birth services at a Thai hospital would vary depending on type of services required ie. c-section vs. uncomplicated birth. Costs for the same services also vary between hospitals. For this reason, we are unable to provide an approximate/average cost.

4. Methods: Why did you choose focus group discussions to investigate this issue rather than interviews with women? What made FGDs the better choice?

Response: We chose focus group discussions as we were seeking group opinion on birth place decision making. Focus groups allow participants the opportunity to have a conversation together about a topic of interest. In addition, several of the authors in our paper had conducted other studies with migrant women in this setting and found that focus groups are better at soliciting input as women prefer to participate in groups rather than individually. 

We have made the following addition on line 131-133 page 6, “In designing the study we opted to utilize focus groups as this method helps to facilitate conversations between participants, is culturally appropriate and has been successfully utilized by other studies with this population.”

5. Methods: Missing information about focus group facilitator: Who facilitated the focus group discussions? One of the authors? An independent facilitator? What was their experience / training / qualifications relevant to conducting FGDs? What were their characteristics? E.g. nationality / gender / languages spoken – could these have impacted how they interacted with participants / how participants interacted with them?

Response: The first author (NT) led the focus group discussion and another author (SN) provided interpretation support. Both have significant training and experience conducting focus group discussions. NT is a Canadian researcher, fluent in English and French who conducted her PhD fieldwork in Thailand. SN is fluent in English, Karen and Burmese and has been integral in implementing multiple community-based studies in the research setting. The section (line 149-157, page 7) now reads, “In the border area NT, a researcher with training in qualitative methods, lead the discussions and SN, a skilled community based researcher, provided interpretation to help facilitate the dialogue. NT speaks English, French and Thai and previously conducted her doctoral fieldwork in Thailand. SN is fluent in English, French, Thai, Karen and Burmese and has provided interpretation and facilitation support for multiple community-based studies in the research setting. In Chiang Mai, Kulyapa Yoonut lead the focus group discussions in Thai and SN provided primary interpretation with support for Shan language from an undergraduate university student. Ms. Yoonut is a Thai study nurse with focus group expertise”.

6. Methods: Missing info about interpreter: Who was the interpreter? Were they independent from the research team or part of research team? Were they known to the participants? Did they have experience / training in facilitating focus groups? What were their relevant characteristics that may have affected dynamic of focus group discussion?

Response: The interpreter SN was a member of the research team. We have provided the following information on her characteristics, “SN is fluent in English, French, Thai, Karen and Burmese and has provided interpretation and facilitation support for multiple community based studies in the research setting” (lines 153-155). A student assisted in translating from Thai to Shan. See previous answer for a full response.

7. How were participants recruited? Approached face to face? Flyers? Told about the study by health workers? Did you record if/how many refused to participate and why?

Response: We have added the following sentence, “Healthcare workers informed potential participants about the study and invited interested persons to meet at a specific time and location to join the discussion.” (lines 175-177, page 8). We did not record the number of participants who refused to participate or reasons for non-participation.

8. Did you sample purposively or use a convenience sample?

Response: We used convenience sampling and have now indicated this on page 8 line 173.

9. What’s the rationale behind your sample size?

We set out to have sufficient participants so that we could have separate group discussions based on gravidity, language and ethnicity. See updated sentence (page 8, line 173-174), “We utilized convenience sampling and where possible we attempted to create focus groups that included women similar in gravidity (primi and multi), language and ethnicity”.

10. Where were focus groups held? (i.e. physical location – heath facilities, community spaces, someone’s home, NGO offices?)

Response: We have added the following clarifying sentence (line 142-147, page 6-7) “All focus groups were held at health facilities in a quiet location, separate from where care was being provided”.

11. Was anyone else present during focus group discussions besides the participants and facilitators?

Response: Yes. See addition, “Members of the research team including CA, AH and a study assistant were present for some of the group discussions” (line 157-159, page 7).

12. Please provide more information about your analysis process, specifically the coding. How many data coders were they? Did they develop a coding tree based on inductive analysis (in order to develop the “emergent themes” you mention in addition to your deductive themes).

Response: See update on Page 8, line 191-193 “Using a coding frame initially comprised of the pre-identified themes, NT coded the data while continuously updating the frame to include emergent themes. Team members subsequently met to discuss the analysis”.

13. Did participants or stakeholder provide feedback on your findings?

Response: Due to the nature of our data collection, that we were collecting data from migrants who are mobile, we were not able to contact them to solicit feedback on our findings. However we met as a team to discuss the research findings. We have updated our methods section as follows (page 8, line 192) “Team members subsequently met to discuss the analysis”. We did however seek feedback from stakeholders, including key informants, on the development of a diagram of pathways to birth registration (Figure 1). This is included in the text, (page 9, lines 205-208) “Based on the data obtained from both focus group participants and key informants we developed a diagram of birth registration pathways (Figure 1). To further triangulate the data we verified our preliminary results with several key informants that we had interviewed as well as additional stakeholders knowledgeable about birth registration.”

14. Who translated the audio files? Was this a member of the research team or an independent translator? Were they involved in the analysis? Was there any process of back translation or process to clarify meaning / identify linguistic nuances / identify cultural assumptions that may occur in the translation process?

Response: An independent translator translated the audio files. We have updated the text (page 8, line 185-188) “Following the focus group discussions, all of the audio files were subsequently transcribed and translated into English by an independent translator.”

We did not do back translation, however a member of our team reviewed some of the audio files in cases where we needed to clarify meaning. We have added the following sentence (page 8, line 186-188), “In the case where we needed to clarify meaning or linguistic nuances, members of the research team fluent in languages used by participants reviewed segments of the audio file”.

15. Results: Much of the information in lines 180-186 would be better presented in a Table, rather than as descriptive text.

Response: We pulled the following sentence from the text “Of the 72 women ANC care was received by 54 at border sites and 18 at hospitals in Chiang Mai; while 27 were in their first pregnancy and 45 were having at least their second pregnancy” and included the information in a newly added table.

16. Line 209: The use of the words ‘significant difference’ here implies a statistical comparison has been performed. Use a synonym for significant, such as notable, meaningful, important.

Response: We have updated this to (page 11, line 255) “notable difference”.

17. Line 282: Please provide USD value of 30,000 baht

Response: Text updated. See page 14, line 330 “990 USD”.

18. Discussion: Women in your study did not consider shared language to be a pressing concern, perhaps because they were mostly assured of being provided services in a language that they spoke, or being provided with a ‘bridging person.’ But is this in contrast or a similarity to other populations internationally? (e.g other migrant populations, indigenous and ethnic minorities) What is in the literature about choice of birth location and shared language /related concepts such as cultural safety?

Response: Thank you for this query. Yes, this experience is different from what we find in the literature on concerns or barriers to care experienced by migrant women in other contexts. We have added a sentence further explaining that in the border region, Thai hospitals frequently have interpreters (page 21, line 524-525), “Thai hospitals in the border region frequently have Karen and Burmese interpreters, although they may have large caseloads”.

We have added the following sentence (pages 21-22, lines 530-532), “In contrast to other settings internationally communication challenges and gaps in healthcare coverage have been documented as difficulties for pregnant migrant women seeking healthcare (3).

We did not find any literature about choice of birth location and share language concepts like cultural safety that is specific to migrants. We hope that our study will help fill this gap.

We have revised our figure and uploaded it into PACE.

---------------- 

Final comments: We thank the reviewers and editor for providing feedback on our manuscript and look forward to the prospect of having it published in PLOS ONE in the near future.

---

## [Decision Letter · Decision Letter 1]

2 Mar 2020

Choosing where to give birth: Factors influencing migrant women’s decision making in two regions of Thailand

PONE-D-19-29338R1

Dear Dr. Jiraporncharoen,

We are pleased to inform you that your manuscript has been judged scientifically suitable for publication and will be formally accepted for publication once it complies with all outstanding technical requirements.

With kind regards,

Calistus Wilunda, DrPH

Academic Editor

PLOS ONE

Additional Editor Comments (optional):

Reviewers' comments:

Reviewer's Responses to Questions

**Comments to the Author**

1. If the authors have adequately addressed your comments raised in a previous round of review and you feel that this manuscript is now acceptable for publication, you may indicate that here to bypass the “Comments to the Author” section, enter your conflict of interest statement in the “Confidential to Editor” section, and submit your "Accept" recommendation.

Reviewer #2: All comments have been addressed

2. Is the manuscript technically sound, and do the data support the conclusions?

Reviewer #2: Yes

3. Has the statistical analysis been performed appropriately and rigorously? 

Reviewer #2: N/A

4. Have the authors made all data underlying the findings in their manuscript fully available?

Reviewer #2: No

5. Is the manuscript presented in an intelligible fashion and written in standard English?

Reviewer #2: Yes

6. Review Comments to the Author

Reviewer #2: Thank you for your thoughtful consideration of my comments, I think the changes that you have made have considerably strengthened the paper, particularly the Methods. I have nothing further to add.

7. PLOS authors have the option to publish the peer review history of their article (what does this mean?). If published, this will include your full peer review and any attached files.

Reviewer #2: No

---

## [Editor Report · Acceptance letter]

4 Mar 2020

PONE-D-19-29338R1 

Choosing where to give birth: Factors influencing migrant women’s decision making in two regions of Thailand 

Dear Dr. Jiraporncharoen:

I am pleased to inform you that your manuscript has been deemed suitable for publication in PLOS ONE. Congratulations! Your manuscript is now with our production department. 

With kind regards,

on behalf of

Dr. Calistus Wilunda 

Academic Editor

PLOS ONE